# Relationship between Subjective Health, the Engel Coefficient, Employment, Personal Assets, and Quality of Life for Korean People with Disabilities

**DOI:** 10.3390/healthcare11222994

**Published:** 2023-11-19

**Authors:** Kyung-A Sun, Joonho Moon

**Affiliations:** 1Department of Tourism Management, Gachon University, Seongnam 13120, Republic of Korea; kasun@gachon.ac.kr; 2Department of Tourism Administration, Kangwon National University, Chuncheon 24341, Republic of Korea

**Keywords:** quality of life, subjective health, the Engel coefficient, employment, personal assets, people with disability

## Abstract

The aim of this research is to examine the effect of subjective health on the quality of life of Korean people with disabilities. The second goal of this study is to examine the effect of the Engel coefficient on quality of life. Additionally, this study is conducted to inspect the effect of employment and personal assets on quality of life. Further, in this work, the moderating effect of personal assets on the association between employment and quality of life for people with a disability is explored. The Panel Survey of Employment for the Disabled served as the source of data. The study period ranges from 2016 to 2018. To test the research hypotheses, this study adopted econometric analyses, namely, ordinary least squares, fixed effect, and random effect models. The results revealed that the quality of life for people with disabilities is positively influenced by subjective health, employment, and personal assets. In contrast, the Engel coefficient exerts a negative impact on quality of life. Plus, the finding indicates that personal assets negatively moderate the relationship between employment and quality of life for people with disabilities. This research is aimed at presenting policy implications for the welfare of people with disabilities.

## 1. Introduction

The definition of a person with disabilities in Korea is an individual who is limited in social life due to both mental and physical disabilities [1]. According to Statistics Korea [2], the number of disabled Koreans is approximately 2.65 million, and the population size of disabled Koreans has been increasing steadily since 2018. Such growth indicates that the proportion of disabled individuals in Korea has increased. Hence, policy design for people with disabilities is likely to become an even more imperative issue in Korean society because the welfare budget for people with disabilities is constrained. To provide a guideline for policy making aimed at people with disabilities, this research is conducted to inspect the characteristics of people with disabilities. The dependent variable of this work is quality of life. Scholars have defined quality of life as an indicator of individual welfare and happiness [3,4]. Numerous studies have also scrutinized the influential attributes on quality of life [5,6,7,8]. Such bountiful works implied that investigating the characteristics of quality of life is valuable. Additionally, quality of life is likely to function as an attribute when determining the characteristics of the disabled.

In this research, the determinants of quality of life were identified as subjective health, the Engel coefficient, employment, and personal assets. Previous studies have uncovered the significant and positive effect of subjective health on quality of life, arguing that healthy conditions are indispensable for improving quality of life [9,10,11]. This study is thus aimed at ensuring the effect of subjective health on quality of life in the domain of people with disabilities. Next, this study inspects the impact of the Engel coefficient on quality of life. The Engel coefficient serves as evidence of poor life conditions because a higher Engel coefficient indicates an insufficient budget for other areas of life, such as hobbies, recreation, education, and medical services [12,13,14]. Despite this argument, few studies have demonstrated the effect of the Engel coefficient on quality of life in the case of individuals with disabilities. This research is aimed at minimizing such a research gap by presenting empirical evidence. The third area of this research is employment. The extant literature has shown that employment plays a significant role in ensuring a better life because it satisfies human needs, such as social needs, achievement needs, esteem, and financial gain [15,16,17]. From this perspective, this study examines the effect of employment on the quality of quality of life of people with disabilities. In addition, this research uses personal assets as a determinant of quality of life because scholars contended that wealth is an essential element in improving individual quality of life [18,19]. Furthermore, this research tests the moderating effect of personal assets on the relationship between employment and quality of life. Labor might exert varied effects on individual life depending on the relevant wealth condition. Specifically, earning from labor is likely to be more desperate for poor individuals than for wealthy individuals because it is directly linked with survival. Namely, the meaning of labor could vary depending on the condition of wealth, and personal assets enable the approximation of individual wealth conditions in this study. The moderating effect of wealth, which clarifies the meaning of labor in the case of Korean people with disabilities, is also scrutinized.

This research sheds light on the literature by elucidating the moderating effect of personal assets on the association between employment and quality of life. This is because prior works exploring the effects of disabilities have rarely assessed such an effect. Moreover, this research is valuable in that it presents empirical evidence on the effects of subjective health, the Engel coefficient, and employment on the quality of life of the disabled. Such efforts might enable confirming the findings in the extant literature. The outcomes of this work are likely to lead policy makers to design a more adequate welfare system for people with disabilities.

## 2. Review of the Literature and Hypothesis Development

### 2.1. Quality of Life

Prior studies have addressed the fact that quality of life is an individual evaluation of individual’s current life conditions and their level of satisfaction with their living condition [3,4,5]. Moreover, prior studies addressed that quality of life reflects the overall satisfaction of individual living [20,21]. Numerous works have adopted quality of life as the main attribute when determining individual behavior. Guida and Carpentieri [22] explored the quality of life using Milan city residents. Zhang and Ma [8] inspected the quality of life by employing Chinese participants. Ravens-Sieberer et al. [7] when researching German primary school students, revealed the antecedents of quality of life. Alsubaie et al. [23] and Berdida and Grande [24] examined influential attributes of quality of life for university students. Uysal and Sirgy [25] investigated the characteristics of the life quality of travelers. Additionally, Moon et al. [26] used quality of life as a dependent variable to scrutinize the behavior of Korean elderly individuals. Moreover, many prior studies have empirically explored the characteristics of people with a disability using quality of life as a principal indicator [27,28,29]. Specifically, Steptoe and Di Gessa [28] inspected the determinants of quality of life by employing people with physical disabilities; Verdugo et al. [29] implemented empirical research to identify the determinants of quality of life by studying people with intellectual disabilities. Given the numerous studies on quality of life, one can see that quality of life is an imperative attribute in various domains.

### 2.2. Subjective Health and Quality of Life

Subjective health is the degree of mental and physiological healthiness assessed from one’s own point of view [9,11,30]. Subjective health is an essential aspect of better living because healthiness is a precondition for improving individuals’ lives [31,32]. A vast body of the literature has empirically demonstrated the effect of subjective health on quality of life. For instance, Qazi et al. [33] showed the positive impact of subjective health on quality of life by researching older women. Miyakawa and Hamashima [34], while researching high school students, suggested that subjective health exerted a positive effect on quality of life. Moon et al. [26] also found a positive association between subjective health and quality of life for Korean senior citizens. In addition, Low et al. [35] explored older people in 20 countries, and their results documented the fact that quality of life is positively affected by subjective health. Next, Skevington et al. [36] examined cross-cultural data, and the results indicated that subjective health is a significant antecedent for quality of life. Also, Kim et al. [10] exposed a significant and positive link between the subjective health of students and their quality of life. Plus, Heyne et al. [37] found that the quality of life for patients is significantly influenced by subjective health conditions. From the literature review, it can be inferred that subjective health is likely to become an essential attribute in assessing individual quality of life. Therefore, the following hypothesis is proposed:

**Hypothesis** **1.**
*Subjective health positively impacts quality of life.*


### 2.3. Engel Coefficient and Quality of Life

The Engel coefficient refers to the proportion of food cost to the total cost of living [6,13]. A low Engel coefficient means that individuals possess more surplus money to make life better because food expenditure is an opportunity cost that coexists with other living-related budgets [14,38]. Ma et al. [39] alleged that the Engel coefficient lowers the overall living quality because of the constrained resource conditions. Yang et al. [13] and Xie et al. [12] also argue that a high Engel coefficient reflects significantly worse individual living conditions because a high proportion of food costs causes resource constraints and increases the likelihood of diseases such as obesity, diabetes, high blood pressure, and heart attack. Also, Shi et al. [40] and Qin et al. [41] showed that the Engel coefficient negatively impacted on quality of life by inspecting the Chinese population. Moon et al. [6] revealed that a higher Engel coefficient reflects a negative impact on the quality of life of Korean elderly individuals. Furthermore, regarding the literature review, it is presumed that a higher Engel coefficient is likely to reflect a more negative quality of life. Thus, the following hypothesis is proposed:

**Hypothesis** **2.**
*A higher Engel coefficient reflects a poorer quality of life.*


### 2.4. Employment and Quality of Life

Prior research has documented the fact that employment is the status of working with economic value [42,43,44]. Working is an instrument for improving quality of life because it provides financial rewards and opportunities to satisfy higher levels of needs, such as social needs, esteem, and self-actualization [15,16,45]. Many studies have shown the effect of employment on quality of life. For example, Carlier et al. [46], when examining Dutch individuals, indicated that quality of life is enhanced by labor. Blalock et al. [47] also documented that employment played an essential function in the improvement of life quality when studying patients with serious diseases. In a similar vein, Kim and Feldman [48] revealed a positive relationship between quality of life and employment by researching retired people. Kober and Eggleton [44] implemented research by employing persons with disability, the findings documented that quality of life is improved by employment. Plus, Beyer et al. [49] disclosed that the quality of life for people with intellectual disabilities was enhanced by employment. Cocks et al. [17] found the significant impact of employment on the living quality of people with disabilities because of various elements: benefits from work, social connection, and job satisfaction. Additionally, previous studies have shown that the employment of people with disabilities is an imperative attribute for the enhancement of their life quality [17,50]. With respect to the literature review, the following hypothesis is proposed:

**Hypothesis** **3.**
*Employment positively impacts quality of life.*


### 2.5. Personal Assets and Quality of Life

Wealth is indispensable for living because wealth allows individuals to consume products and services [18,19,51]. Wealthier people are more likely to consume better-quality services and goods because they have greater financial power [52,53,54]. Luburić and Fabris [20] contended that wealth is a critical element for better living. Diwan [18] also claimed that wealth plays a significant attribute in a better life because it can cause emotional security in an individual. An et al. [55] uncovered the fact that financial status is positively associated with quality of life. Xiao et al. [56] showed that students’ quality of life is positively influenced by financial power. Zafar et al. [19] conversely demonstrated that financial burden exerted a negative impact on quality of life. Regarding the literature review, the following research hypothesis is proposed:

**Hypothesis** **4.**
*Personal assets positively impact quality of life.*


### 2.6. Moderating Effect of Personal Assets on the Relationship between Employment and Quality of Life

The law of diminishing marginal utility posits that individuals gain a higher level of utility in consumption when their resources are scarce [57,58]. Kober and Eggleton [44] demonstrated that the quality of life is influenced in varied manners by the situation of workers by scrutinizing people with disability. Gautié and Schmitt [59] documented that the meaning of labor appeared in different manners depending on the wealth condition because labor is compulsory for people with lower levels of wealth. Also, labor functions to provide the money necessary for living [43,44], and this resource is likely to exert a stronger impact on quality of life in the case of individuals with fewer assets, according to the law of diminishing marginal utility [60,61]. Namely, the impact of labor is likely to vary depending on the relevant wealth condition. Thus, the following hypothesis is proposed:

**Hypothesis** **5.**
*Personal assets negatively moderate the relationship between employment and quality of life.*


## 3. Methods

### 3.1. Research Model and Data Collection

Figure 1 exhibits the research model. The dependent variable is quality of life. Quality of life is positively impacted by subjective health. The Engel coefficient negatively affects quality of life. Employment is also positively associated with quality of life. Additionally, personal assets exert a positive effect on quality of life, and personal assets further moderate the relationship between labor and quality of life.

The data collection was implemented using the Panel Survey of Employment for the Disabled, which was published by the Employment Development Institute. The Panel Survey of Employment for people with disabilities provides researchers with survey information about people with disabilities. The study period spanned from 2016 to 2018. After 2018, the Employment Development Institute did not continue to report such survey information. The data appeared as panel data which consists of multiple periods and times [62]. Panel data refer to data that references multiple participants and multiple time points. In the case of this study, the data appeared as unbalanced panels because all participant information was not matched throughout the entire study period [62,63]. The initial observation of this research was 12,225. In the data cleaning process,836 observations were eliminated because of no response in the dataset. Thus, the total number of valid observations for data analysis was 11,389.

### 3.2. Variable Illustration and Analytic Instrument

Table 1 describes the measurements of the variables. Quality of life (QL) was measured with a five-point scale (from 1 = very dissatisfied to 5 = very satisfied). The question for QL is ‘how are you satisfied with your living?’ The measurement of subjective health (SH) is a four-point scale (from 1 = very poor to 4 = very good). The question is ‘how do you assess your overall health condition?’. The Engel coefficient is measured by monthly food expenditure over monthly total living expenses. Employment (EM) appeared as a binary variable (0 = unemployed, 1 = employed). Personal asset (PA) was measured by a natural log of the amount of wealth possessed by survey participants, and its unit was 10,000 KRW. With regard to the control variables, disability level (DL) (0 = mild, 1 = severe) and gender (GN) (0 = male, 1 = female) were both measured as dummy variables. Age was the physical age (AG) of the survey participant.

First, descriptive statistical analysis was conducted to compute the basic information of the study variables. A Spearman correlation matrix was chosen to examine the correlation coefficients between variables. To test the hypotheses, three econometric instruments, namely, ordinary least squares, fixed effects model, and random effects model, were applied. Ordinary least squares are a regression model that minimizes the sum of squared residual in the estimation [62,64]. The fixed effect model incorporates multiple time-related variables in the form of dummy variables into the model for the minimization of omitted variable bias in relation to time. Next, the random effect model refers to a regression model that contains unobservable effects in the estimation [63,64]. To test the moderating effect, two variables were multiplied, EM and PA (EM × PA). Then, the EM × PA variable was incorporated into the regression model. Furthermore, median split analysis (personal assets median value = 6000) was used to further scrutinize the direction of the moderating effect. Moreover, this work used three control variables (e.g., gender (GN), age (AG), and disability level (DL)) in the regression model. Plus, this research implemented the Lagrangian multiplier test and Hausman test to select the best model for panel data estimation. The results of both tests were not statistically significant. It suggests that ordinary least squares are the most appropriate econometric instrument for parameter estimation [62,63]. Moreover, this research performed ordinal logit analysis for the robustness check because it is more lenient for the normality assumption [62,63]. The regression equation model is as follows:QL_it_ = β_0_ + β_1_ SH_it_ + β_2_ EG_it_ + β_3_ EM_it_ + β_4_ PA_it_ + β_5_ EM × PA_it_ + β_6_ GN_it_ + β_7_ AG_it_ + β_8_ DL_it_ + ε_it_(1)
where ε is the residual, i is the ith participants, t is the tth year, QL is quality of life, SH is subjective health, EG is the Engel coefficient, EM is Employment, PA is personal assets, GN is gender, AG is physical age, DL: disability level.

## 4. Main Findings and Discussion

### 4.1. Descriptive Statistics and Correlation Matrix

Table 2 illustrates the descriptive statistics. The number of observations is 11,389. The mean values (standard deviation) of QL, SH and EG are 3.27 (SD = 0.69), 2.48 (SD = 0.66), and 0.31 (SD = 0.13), respectively. Fifty percent of the survey participants were employed, as presented in Table 2. The mean and standard deviation of PA are 13,787.44 and 26,001.21, respectively. Table 2 also presents the information for GN (mean = 0.65, SD = 0.47), AG (mean = 43.71, SD = 12.60), and DL (mean = 0.32, SD = 0.46). The mean value of AGE is 43.71, and its standard deviation is 12.60, with minimum and maximum values of 15 and 66, respectively.

Table 3 exhibits the spearman correlation matrix. QL is positively correlated with SH (r = 0.429, *p* < 0.05), EM (r = 0.343, *p* < 0.05), PA (r = 0.348, *p* < 0.05), and GN (r = 0.036, *p* < 0.05). However, QL negatively correlated with EG (r = −0.232, *p* < 0.05), AG (r = −0.122, *p* < 0.05), and DL (r = −0.165, *p* < 0.05). Next, SH is positively correlated with EM (r = 0.367, *p* < 0.05), PA (r = 0.281, *p* < 0.05) and GN (r = 0.096, *p* < 0.05), while SH is negatively correlated with EG (r = −0.228, *p* < 0.05), AG (r = −0.266, *p* < 0.05), and DL (r = −0.140, *p* < 0.05). Additionally, EG is negatively correlated with EM (r = −0.295, *p* < 0.05) and PA (r = −0.355, *p* < 0.05, whereas it is positively correlated with AG (r = 0.102, *p* < 0.05) and DL (r = 0.156, *p* < 0.05).

### 4.2. Results of Hypotheses Testing

Table 4 shows the results of testing the hypotheses. The dependent variable is QL. All three econometric models are statistically significant regarding F-values and Wald χ^2^ (*p* < 0.05). SH exerts a positive effect on QL (β = 0.345, *p* < 0.05). EG negatively affects QL (β = −0.338, *p* < 0.05), whereas EM positively impacts QL (β = 0.270, *p* < 0.05). It also appears that QL is positively influenced by PA (β = 3.47 × 10^−6^, *p* < 0.05). Moreover, the moderating variable EM × PA (β = −8.14 × 10^−7^, *p* < 0.05) exerts a negative effect on QL. Namely, PA negatively moderates the association between EM and QL. The results are consistent in all three models in terms of significance and direction. Therefore, all the proposed hypotheses are supported.

Table 5 presents the results of testing the hypotheses using ordinal logit. The dependent variable is QL. All models are statistically significant regarding F-values and Wald χ^2^ (*p* < 0.05). SH exerts a positive effect on QL (β = 1.136, *p* < 0.05). EG negatively affects QL (β = −1.171, *p* < 0.05), whereas EM positively impacts QL (β = 0.860, *p* < 0.05). It also appears that QL is positively influenced by PA (β = 1.36 × 10^−5^, *p* < 0.05). The results are consistent in all models in terms of significance and direction.

Table 6 depicts the additional analysis to inspect the moderating effect of personal assets. The mean values of the high assets group and unemployed and that of the high assets group and employed are 3.26 and 3.61, respectively. Additionally, the mean value of the low assets group and the unemployed group is 2.88 with 0.70 as the standard deviation. Finally, the mean value of the low assets and the employed group is 3.34. Additionally, Figure 2 presents a graphical illustration of the moderating effect of personal assets.

## 5. Conclusions

The purpose of this study is to examine the determinants of quality of life in people with disabilities in Korea. The results show that subjectively healthier participants are better in their lives. This implies that subjective health becomes a precondition for a better quality of life for people with disabilities in Korea. The findings of this research appeared similar compared with the research implemented by employing an international sample [35] and performing cross-cultural research [36]. It can be inferred that the positive effect of subjective health on quality of life is also crucial in the case of Korean people with disabilities. Moreover, the findings document the fact that the Engel coefficient reflects a negative effect on the quality of life for people with disability in Korea. Specifically, food costs could become a burden for people with disabilities from the perspective of achieving a higher quality of life. The results displayed a similar pattern in the Korean case as compared to the case researched by the Chinese population [40,41]. The results also reveal that employed participants had a better quality of life than unemployed participants. Namely, this suggests that employment plays an essential role in improving the quality of quality of life of people with disabilities. Additionally, the results indicate that wealth is a crucial attribute for a better quality of life for people with disabilities in Korea.

The crux of this work is the moderating effect of personal assets on the relationship between employment and quality of life. With regard to the moderating effect, it was found that the group with fewer assets was more sensitive to labor than the group with greater assets. That is, the gap between employment and unemployment was larger in the case of possessing fewer assets. Such a fact may be the reason for the negative moderating effect of personal assets on the relationship between employment and quality of life. It can also be inferred that labor might become more important in the case of those with disabilities who possess fewer assets for elevating quality of life. In other words, the value of labor could be varied whether it is compulsory or voluntary. The high-level personal assets group is more likely to become a voluntary group, whereas the low-level personal assets group is more likely to be a compulsory group. Given the moderating effect of personal assets, it could be inferred that working might become a necessary means of survival for the compulsory group.

From the results of including the control variables, it can be inferred that quality of life decreased for males, older individuals, and those with severe disabilities. It is aligned with the findings of previous research. In detail, Shim et al. [65] showed that people with severe disability showed worse appraisal for their living because their life is more constrained than people with mild disabilities. The findings of the extant literature also support the findings of this work in that individual quality of life varies by gender and age [26,27,65,66].

This research contributes to the literature by ensuring the moderating effect of personal assets on the association between employment and quality of life. Because labor plays a significant role in obtaining funds, the value of labor is likely to vary depending on individual financial conditions. However, few studies have examined the moderating effect of personal assets on the relationship between labor and quality of life. This study is thus aimed at minimizing this research gap, and this work unveils the significant link among variables. This achievement represents the value of this research by elucidating the association between variables. Scholars have argued that the Engel coefficient reflects an important effect on the quality of life of individuals because a higher Engel coefficient indicates a reduction in other living areas aside from food [15,16,26]. Despite such an argument, prior studies have rarely reported empirical evidence for the relationship between the Engel coefficient and the quality of life of individuals with disabilities. Hence, the next contribution of this research is to demonstrate the effect of the Engel coefficient on the quality of life of people with disabilities and thus to fill this research gap. In addition, this study contributes to the literature by documenting the significant impact of subjective health on quality of life, which supports the results of the extant literature [6,10,33]. Thus, this study externally validates the findings of previous studies in terms of the link between subjective health and quality of life.

This work has policy implications. First, government budgets for people with disabilities might need to be allocated to enhance both mental and physiological health conditions. Such budgets can enable investments in medical care, social work, and counseling care services for people with disabilities. Also, government resources could be allotted to outdoor activities for people with disabilities such as leisure and cultural activities because they can refresh the individual mental condition. Moreover, policy makers may be able to consider investing resources in supporting food expenditures because food costs hinder people with disabilities from spending their money on other areas and thus improving their quality of life. This could be accomplished by direct food aid and issuing food coupons for people with disabilities. Next, policy makers need to more efficiently allocate government money to job creation. In detail, policy makers might be able to reduce the unemployment rate of disabled individuals with disabilities by offering jobs with adequate wage levels. This could be accomplished by investigating the type of disabilities that are prevalent in the population because the potential for employment varies across types of disabilities. Such an action would enable policy makers to find ways to use constrained resources in a more effective manner. Furthermore, policy makers might be able to contemplate the resource for poor people with disabilities more because their labor is compulsory, which is likely to degrade the life quality. However, the criteria for detecting financially distressed people needs to be designed in a more delicate manner. Otherwise, the government budget is likely to be allocated inadequately.

This study has some limitations. First, the measurement of labor was a binary measurement. Future research might be able to contemplate various aspects of labor, such as salary level, job stability, and working conditions. This inclusion might be useful for attaining more concrete implications for policy design. Additionally, this study depended on archival data. Future research should consider directly surveying people with disabilities, which could become an avenue for acquiring more robust estimation results using more advanced statistical instruments. Last, the sample of this work is limited to the case of Korea. Future works might be able to consider cases from other nations; ensuring the link between variables could make the results of this study more generalizable.

## Figures and Tables

**Figure 1 healthcare-11-02994-f001:**
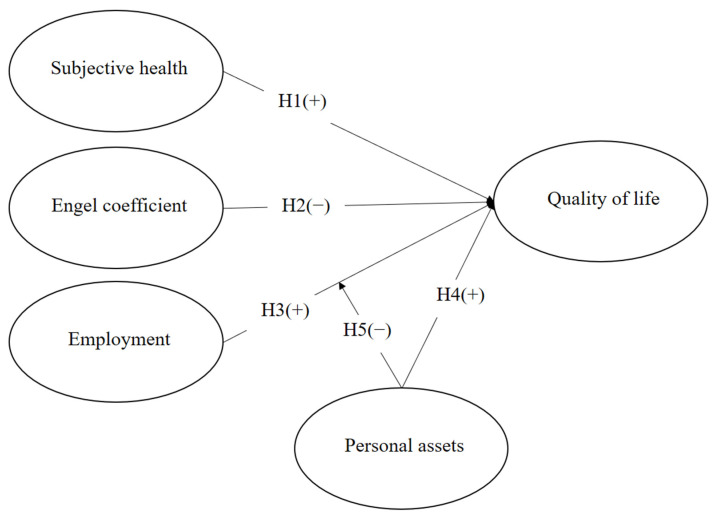
Research model.

**Figure 2 healthcare-11-02994-f002:**
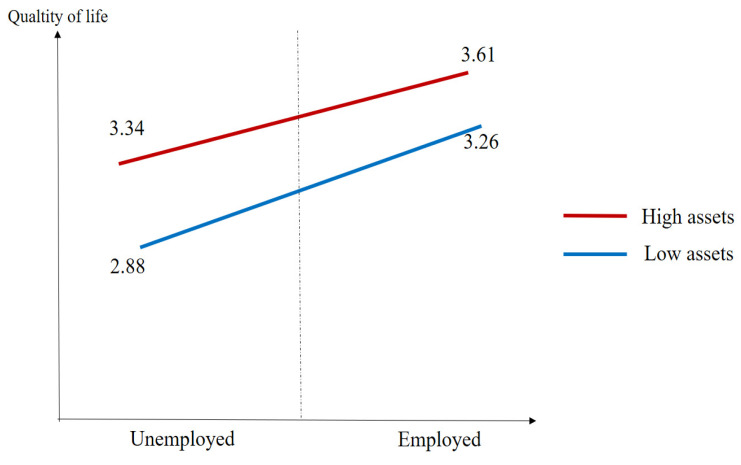
Graphical presentation for the results of moderating effect.

**Table 1 healthcare-11-02994-t001:** Variable description (N = 11,389).

Name	Code	Description (Unit)
Quality of life	QL	(1 = very dissatisfied, 5= very satisfied)
Subjective health	SH	(1 = very poor, 4 = very good)
Engel coefficient	EG	Monthly food expense/Monthly total expense
Employment	EM	(0 = unemployed, 1 = employed)
Personal assets	PA	(Personal assets) (10,000 KRW)
Gender	GN	(0 = male, 1 = female)
Age	AG	Physical age of survey participants
Disability level	DL	(0 = mild, 1= severe)

Note: KRW denotes Korean won.

**Table 2 healthcare-11-02994-t002:** Descriptive statistics (N = 11,389).

Variable	Mean	SD	Minimum	Maximum
QL	3.27	0.69	1	5
SH	2.48	0.66	1	4
EG	0.31	0.13	0	1
EM	0.50	0.50	0	1
PA	13,787.44	26,001.21	0	600,000
GN	0.65	0.47	0	1
AG	43.71	12.60	15	66
DL	0.32	0.46	0	1

Note: SD denotes standard deviation, QL: quality of life, SH: subjective health, EG: Engel coefficient, EM: employment, PA: personal assets, GN: gender, AG: physical age, DL: disability level.

**Table 3 healthcare-11-02994-t003:** Spearman correlation matrix.

Variable	1	2	3	4	5	6	7
1. QL	1						
2. SH	0.429 *	1					
3. EG	−0.232 *	−0.228 *	1				
4. EM	0.343 *	0.367 *	−0.295 *	1			
5. PA	0.348 *	0.281 *	−0.355 *	0.281 *	1		
6. GN	0.036 *	0.096 *	−0.018	0.220 *	0.034 *	1	
7. AG	−0.122 *	−0.266 *	0.102 *	0.017	−0.091 *	−0.067 *	1
8. DL	−0.165 *	−0.140 *	0.156 *	−0.305 *	−0.173 *	−0.041 *	−0.189 *

Note: * *p* < 0.05, QL: quality of life, SH: subjective health, EG: Engel coefficient, EM: employment, PA: personal assets, GN: gender, AG: physical age, DL: disability level.

**Table 4 healthcare-11-02994-t004:** Results of hypotheses testing using linear regression.

Variable	Model 1 (O) β (t-stat)	Model 2 (F) β (t-stat)	Model 3 (R) β (wald)
Intercept	2.502 (61.46) **	2.502 (61.29) **	2.502 (61.46) **
SH	0.345 (35.77) **	0.345 (35.78) **	0.345 (35.77) **
EG	−0.338 (−7.45) **	−0.338 (−7.43) **	−0.338 (−7.45) **
EM	0.270 (18.42) **	0.270 (18.42) **	0.270 (18.42) **
PA	3.47 × 10^−6^ (10.26) **	3.47 × 10^−6^ (10.27) **	3.47 × 10^−6^ (10.26) **
EM × PA	−8.14 × 10^−7^ (−1.83) *	−8.14 × 10^−7^ (−1.84) *	−8.14 × 10^−7^ (−1.83) *
GN	−0.062 (−5.14) **	−0.062 (−5.14) **	−0.062 (−5.14) **
AG	−0.002 (−4.28) **	−0.002 (−4.22) **	−0.002 (−4.28) **
DL	−0.077 (−5.88) **	−0.077 (−5.87) **	−0.077 (−5.88) **
F-value	491.36 *	491.41 *	
Wald χ^2^			3930.88 *
R^2^	0.2579	0.2579	0.2579

Note: Dependent variable: QL, * *p* < 0.1, ** *p* < 0.05, O is ordinary least square, F is fixed effect, R is random effect, QL: quality of life, SH: subjective health, EG: Engel coefficient, EM: employment, PA: personal assets, GN: gender, AG: physical age, DL: disability level.

**Table 5 healthcare-11-02994-t005:** Results of hypotheses testing using ordinal logit.

Variable	Model 4 (OL) β (t-stat)	Model 5 (OF) β (t-stat)
SH	1.136 (32.95) **	1.137 (32.96) **
EG	−1.171 (−7.66) **	−1.171 (−7.63) **
EM	0.860 (16.73) **	0.860 (18.42) **
PA	1.36 × 10^−5^ (9.31) **	1.37 × 10^−5^ (9.34) **
EM × PA	−1.59 × 10^−6^ (−0.86)	−1.61 × 10^−6^ (−1.87)
GN	−0.194 (−4.73) **	−0.194 (−4.74) **
AG	−0.006 (−4.10) **	−0.006 (−4.02) **
DL	−0.244 (−5.53) **	−0.243 (−5.51) **
LR χ^2^	3319.91 *	3321.56 *
Psuedo R^2^	0.1408	0.1409

Note: Dependent variable: QL, * *p* < 0.1, ** *p* < 0.05, OL is ordinal logit, OF is ordinal logit with fixed effect, QL: quality of life, SH: subjective health, EG: Engel coefficient, EM: employment, PA: personal assets, GN: gender, AG: physical age, DL: disability level.

**Table 6 healthcare-11-02994-t006:** Results of moderating effect of personal assets.

Variable	Unemployed Mean (SD)	Employed Mean (SD)
High-assets group	3.26 (0.66)	3.61 (0.55)
Low-assets group	2.88 (0.70)	3.34 (0.61)

Note: Dependent variable: QL.

## Data Availability

Data are contained within the article.

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
