# Peer review of "Relationship between Subjective Health, the Engel Coefficient, Employment, Personal Assets, and Quality of Life for Korean People with Disabilities"

_healthcare, 2023, doi:10.3390/healthcare11222994_

Round 1

Reviewer 1 Report

Comments and Suggestions for Authors

The authors are to be commended for their efforts in this study, which examined the impact of subjective health on the quality of life of people with disability in Korea, as well as the impact of Engel's coefficient on quality of life.

I would like to request the revision for the authors in the following points.

1. The definition of QOL.

The authors rate QOL on a satisfaction scale of 1-5.

In general, research studies of QOL have included the following components, as found in the WHO QOL questionnaire: physical health, Psychological, Social relationships and Environment domains. On the other hand, many QOL indicators have items that ask about physical conditions, which makes them not useful tools for people with disabilities.

In the literature on QOL cited by the authors, it is necessary to check what definitions are used and whether they are the same as those defined by the authors in the present study.

In view of the above, the authors should be cautious about using the term QOL in this paper.

Instead of QOL, the term might be better to amend to "Satisfaction for daily life".

2. Method - analysis

1) I’d like to request the authors to use categorical data for personal assets in the final model. Its SD is large and the variability of the data is a concern.

3,Discussion

 There are five research hypotheses, but these are looking at the association with 'satisfaction with quality of life', which we believe has already been evaluated in a large body of literature. A noteworthy aspect of this study is that it targets people with disabilities. In the discussion, little literature review has been conducted. Additional discussion is requested with reference to the following points.

(L275)

The authors discussed that quality of life decreased for males, older  individuals, and those with severe disabilities, however, no referred the other literatures pros and cons on this. Please add this to discussion.

(L-260 )

“This implies that subjective health becomes a precondition for a better quality of life for people with disabilities in Korea. Moreover, the findings document the fact that the Engel coefficient reflects a negative effect on the quality of life for the disabled in Korea.”

There is no other evidences referred on this. If it is difficult for finding among people with disability, please add any other arguments in other countries for international audiences. Also, please describe what specific policies are desired.

Reviewer 2 Report

Comments and Suggestions for Authors

The paper covers an interesting topic. Quality of life of persons with disabilities is an important aspect and not so deeply studied so far as quality of life in general. The article investigates the impact of chosen factors on quality of life of PWD, particularly the impact of subjective health, Engel coefficient, employment and personal assets. The analysis concerns Korea and is based on data from a Panel Survey of Employment for the Disabled giving the sample 11 389. The strengths of the work are: the problem under discussion, statistical/econometric analysis performed on a big sample from a national survey, examining the moderating effect of personal assets on the relationship between labor and quality of life. The results may have policy implications.

Although interesting and having certain advantages, the article contains some shortcomings and some issues are not sufficiently explained or specified:

1.      Who is considered people with disabilities in Korea? As the understanding and definitions of persons with disabilities vary across countries and researchers it should be explained who was taken into account in this study.

2.      Line 16: it should be “squares” not “square”. The same in description in Table 4.

3.      You use terms “disabled” and “people with disability” interchangeably. Is it intentional? Consider using persons or people with disability (PwD) throughout your paper as it seems to be a preferred expression nowadays.

4.      Line 65: it should be “disabilities” not “Disabilities”.

5.      Lines 76-77: “quality of life is an individual evaluation of individual’s current life conditions and their level of satisfaction with their quality of life”. This should be reformulated. Quality of life cannot be explained by quality of life.

6.      Line 136: “in retired people”. Is “in” appropriate here?

7.      Line 171: “The data were implemented using”. What “implemented” means in this sentence?

8.      Line 179: “total number of valid observations”. What are valid observations for this study? Please explain.

9.      Table 1: QL and SH have more than two levels which is suggested in the last column. Better write for QL (from 1 = Very dissatisfied to 5= Very satisfied) or similarly. The same for SH.

10.   Line 195: The word “model” is missing after “fixed effects, and random effects”.

11.   Lines 195-196: “Ordinary least squares is a regression model that minimizes the residual in the estimation”. In fact it minimizes the sum of squared residuals. Please correct.

12.   You use three models, e.g. ordinary least squares regression, fixed effects model and random effects model but you present only one equation after the line 206 which seems to correspond to the classical approach. The fixed effects model and random effects model are more sophisticated and you don’t give the explanation how you specified them. The second thing is that you don’t provide the justification for using those three types of models. In my opinion the explanation why and how the models were constructed and estimated is necessary. I strongly advise checking the model specifications and calculations once again (especially since the results obtained are very similar). Additionally, the Hausman test is often used to compare fixed and random effects models and gives a clue to point out a better model. That may help you to enrich your approach. To conclude: describe how models were specified and estimated, justify usage of them (if all three models are necessary in your research and why), check the results.

13.   Lines 204-205: “Moreover, three control variables (e.g., gender (GN), age (AG),and disability level (DL)) in the regression model.” Something is missing in this sentence.

14.   Line 236: “All three econometric models are statistically significant (p<.05).”. What do you mean by this? This statement is too general. Please specify what you mean (significance of model parameters?) and if so what tests prove this.

15.   Figure 2: When illustrating the moderating effects usually the IV is put on the x-axis, DV on the y-axis and different lines are for different levels of the moderator. In your figure it is just opposite – you put moderator on the X-axis. Consider rearranging the figure and adding the comment on the position of the lines.

16.   Lines 259-260 “The results show that both mentally and physiologically healthier participants are better in their lives”. I didn’t find in your research the division into mentally and physiologically healthier participants. The variable SH is a categorical one as you described and reflects the subjective health in four-point scale (from 1 = Very poor, to 4 = Very good). So I think that such a conclusion cannot be drawn from your research. Please reformulate this statement.

17.   Line 284: “This achievement represents is the value”. “is” is not appropriate here.

18. According to Instructions for Authors: "References must be numbered in order of appearance in the text (including table captions and figure legends) and listed individually at the end of the manuscript." 

Reviewer 3 Report

Comments and Suggestions for Authors

Thank you for the opportunity to review this article.

The article has several methodological errors, it does not show how the survey was conducted, the statistical results are not as correct as possible and even more so with surveys.

The conclusions should appear in another section

Reviewer 4 Report

Comments and Suggestions for Authors

The manuscript was written inadequate. The authors should follow the journal instruction to prepare the manuscript, from title to discussion and references. The manuscript lost structured format that made the content difficult to understand. I suggest the authors can also take a published paper as your reference while you are writing the manuscript.

Comments on the Quality of English Language

poor

Round 2

Reviewer 1 Report

Comments and Suggestions for Authors

The authors have responded satisfactorily to the peer review comments.

The concluding section should be written without references. Texts with references should be added to the discussion.

Reviewer 3 Report

Comments and Suggestions for Authors

Congratulations to the authors for their efforts.

Reviewer 4 Report

Comments and Suggestions for Authors

The authors did not address all of my comments.

Comments on the Quality of English Language

Acceptable
